# Sequencing of the Complete Mitochondrial Genome of *Pingus sinensis* (Spirurina: Quimperiidae): Gene Arrangements and Phylogenetic Implications

**DOI:** 10.3390/genes12111772

**Published:** 2021-11-08

**Authors:** Fanglin Chen, Hong Zou, Xiao Jin, Dong Zhang, Wenxiang Li, Ming Li, Shangong Wu, Guitang Wang

**Affiliations:** 1College of Science, Tibet University, Lhasa 850000, China; chenfanglin@ihb.ac.cn; 2Key Laboratory of Aquaculture Disease Control, Ministry of Agriculture and State Key Laboratory of Freshwater Ecology and Biotechnology, Institute of Hydrobiology, Chinese Academy of Sciences, Wuhan 430072, China; zouhong@ihb.ac.cn (H.Z.); jinxiao360@gmail.com (X.J.); liwx@ihb.ac.cn (W.L.); liming@ihb.ac.cn (M.L.); wusgz@ihb.ac.cn (S.W.); 3University of Chinese Academy of Sciences, Beijing 100049, China; 4State Key Laboratory of Grassland Agro-Ecosystem, Institute of Innovation Ecology, Lanzhou University, Lanzhou 730000, China; dongzhang0725@gmail.com

**Keywords:** mitogenome, gene order, *Pingus sinensis*, Quimperiidae, spirurina, phylogeny

## Abstract

Despite several decades of intensive research on spirurine nematodes, molecular data on some of the main lineages are still absent, which makes taxonomic classification insufficiently resolved. In the present study, we sequenced the first complete mitogenome for the family Quimperiidae, belonging to *P. sinensis* (Spirurina: Quimperiidae), a parasite living in the intestines of snakehead (*Ophiocephalus argus*). The circular mitogenome is 13,874 bp long, and it contains the standard nematode gene set: 22 transfer RNAs, 2 ribosomal RNAs and 12 protein-coding genes. There are also two long non-coding regions (NCR), in addition to only 8 other intergenic regions, ranging in size from 1 to 58 bp. To investigate its phylogenetic position and study the relationships among other available Spirurina, we performed the phylogenetic analysis using Bayesian inference and maximum likelihood approaches by concatenating the nucleotide sequences of all 36 genes on a dataset containing all available mitogenomes of the suborder Spirurina from NCBI and compared with gene order phylogenies using the MLGO program. Both supported the closer relationship of Ascaridoidea to Seuratoidea than to Spiruroidea. Pingus formed a sister-group with the Cucullanus genus. The results provide a new insights into the relationships within Spirurina.

## 1. Introduction

Nematoda, also known as roundworms, is one of the largest phyla in the animal kingdom. They are pseudocoelomic animals, mostly with small, cylindrical bodies. Nematodes are among the most common, abundant and ecologically diverse animal groups. Free-living species inhabit almost every environment, and they are extremely abundant in soils and aquatic sediments, both freshwater and marine [1,2]. The phylogeny of nematodes depends on a limited number of morphological and ecolog-ical characteristics, which can be determined before using molecular data. Due to the extreme speciosity of this phylum, the poor resolution conferred by morphologic characters resulted in incongruent classification systems [3]. Thus, the identification and phylogeny of nematodes are more suitable used with molecular data. However, since 18S rRNA genetic marker has insufficient resolution for such a large phylum, further studies must employ markers with much higher resolution power. [4]. Complete mitogenomes are much better suited for the taxonomic identification and phylogenetic studies of nematodes than morphology or traditionally used molecular markers because of much higher resolution power [5,6,7,8].

Mitochondrial gene order, i.e., the order in which genes are arranged on the mitochondrial genome, is sometimes used for phylogenetic reconstruction [9,10] and confirmation of findings based on mitochondrial DNA sequences [11]. As a result, mitogenomic data have often proven capable of clarifying phylogenetic relationships where other types of data offered a poor resolution, or where there was discordance between morphological features and genetic data, Furthermore, in many metazoans, the mitogenome accumulates mutations rate faster than the nuclear genome, which makes mitogenome a reliable source of phylogenetic data for distinguishing closely related taxa [12,13]. While mitogenomic data also have limitations for evolutionary studies [14,15], mitochondrial phylogenomics and comparative mitogenomics (e.g., gene order) have often been used as a useful tool for clarifying phylogenetic relationships in a large number of diverse animal groups with their ancient evolutionary origin [16,17,18,19]. Further comparative analyses of major nematode clades poorly represented in terms of mitogenomes are needed to fully explore the mitogenomic gene order evolution.

Spirurina, the representative suborder of gastrointestinal or tissues parasites of all clades of vertebrates, contains infraorders Ascaridomorpha, Gnathostomatomorpha, Oxyuridomorpha, Rhigonematomorpha, Spiruromorpha and Dracunculoidea as *incertae sedi* in the recent classification system based on molecular data [3]. Although the Spirurina clade is stably supported by the SSU rRNA data [20,21], the phylogenetic relationships for Spirurina based on mitogenomes remain inconclusive, largely due to the lack of sequences from several key lineages, which confuses the relationships within the clade and with other clades [22]. *P. sinensis* (Spirurina: Quimperiidae) is a roundworm parasitizing the intestine and caecum of fishes of the genus *Channa*. Although its morphological characteristics are unambiguous, molecular data is still absent, so its molecular phylogeny remains unstudied. As there are currently no complete mitogenomes available for the family Quimperiidae (Chromadorea: Spirurina: Seuratoidea), this study aims to sequence and characterize the complete mitogenome of the Quimperiidae species *P. sinensis*. As there are currently no molecular records at all for this species, these data shall also facilitate the identification of this species for future studies. Another objective was to conduct comparative analyses with all available Spirurina mitogenomes to study the relationships of the Spirurina. In addition, we also conducted comparative analyses of gene orders among the Spirurina, with the goal of inferring the evolutionary history of gene orders. The study first reveals phylogenetic position of family Quimperiidae and advances our understanding of the relationships within Spirurina.

## 2. Materials and Methods

### 2.1. Specimen Collection

Parasitic nematodes were acquired post mortem from the intestine of snakehead (*Channa argus*) samples caught by fishermen in the Lake Xiannv in Xinyu city, Jiangxi Province (24°29′14″–30°04′41″ N; 113°34′36″–118°28′58″ E), China, on 5 August 2020 and rinsed several times with physiological saline to wash out tissue particles of the host. The parasite was identified morphologically under a stereomicroscope and a light microscope according to the morphological features described by Moravec [23]. To further confirm the identity using molecular data, we relied on the *18S* gene, which is recognized as a suitable molecular tool for the identification of nematodes [24]. The gene was amplified using self-designed primers. The sequence exhibited a 97.42% similarity with *Paraseuratum* sp. (Spirurina: Quimperiidae) (GenBank: KP275686). The samples were stored in 99% ethanol and 4 °C.

### 2.2. DNA Extraction and Amplification

The DNA extraction from a single specimen using a TIANamp Genomic DNA Kit (Tiangen Biotech, Beijing, China), following the manufacturer’s recommended protocol, and stored at −20 °C. Primers were designed to match generally conserved regions of target genes and used to amplify short fragments of *cox1, rrnL*, *nad5*, *rrnS*, *nad1*, *cytb* and *nad4* genes. Specific primers were designed based on conserved regions and used to amplify the remaining mitogenome sequence in several PCR reactions (Table 1). The total volume for PCR was 20 μL: 1 μL r*Taq* polymerase (250U, Takara), 10 μL 2 × PCR buffer (Mg^2+^, dNTP plus, Takara, Dalian, China), 0.6 μL of each primer, 7.4 μL ddH_2_O and 1 μL of DNA template. The PCR amplification involved in an initial denaturation at 94 °C for 2 min, followed by 35 cycles of denaturation at 94 °C for 30 s, annealing at 55 °C for 30 s, and extension at 72 °C for 1 min/kb; a final extension at 72 °C for 10 min was then accomplished. The PCR products were sequenced bi-directionally at Sangon Company (Shanghai, China) using the primer-walking strategy.

### 2.3. Genome Assembly

After quality-proofing of the obtained fragments, the complete mitogenomic sequence of *P. sinensis* was assembled manually and aligned comparing with mitogenomes of related species by DNAstar v7.1 software [25] in accordance with the methodology outlined by Zou et al. [26] and Zhang et al. [27,28,29]. Briefly, raw mitogenomic sequences were imported into the MITOS web server to conclude the near boundaries of genes. The precise positions of protein-coding genes (PCGs) were inferred by searching for ORFs (employing the genetic code 5, invertebrate mitochondrial) and further corroborated via a comparison with homologs in Geneious [30]. All tRNAs were identified using ARWEN [31], DOGMA [32] and putative secondary structures were predicted using MITOS [33]. The genome annotation, reported in a Word (Microsoft Office) document, was input into PhyloSuite software [34] to parse and abstract the data, as well as generate GenBank submission files and organization tables for the mitogenome. Codon usage and relative synonymous codon usage (RSCU) for twelve protein-encoding genes (PCGs) of *P. sinensis* and *Cucullanus robustus* (Cucullanidae; the only complete mitogenome from the superfamily Seuratoidea currently available in the GenBank) were computed and the RSCU figure drawn using PhyloSuite. Tandem repeats in the non-coding region (NCRs) were identified by Tandem Repeats Finder [35], and the secondary structures of tRNAs were predicted by MITOS. PAL2NAL web tool [36] was used to calculate non-synonymous (dN)/synonymous (dS) mutation rates for twelve PCGs of *P. sinensis* and *C. robustus*. Rearrangement events and pairwise comparisons of gene orders of twelve spirurine nematodes were studied with CREx web tool [37] using the breakpoint comparison measurement. Bright colors represent similar genomes and dark colors represent dissimilar gene orders.

### 2.4. Phylogenetic and Gene Order Analyses

For the phylogeny, all available mitogenomes belonging to the suborder Spirurina were downloaded from the GenBank, and left only one mitogenome per species. Furthermore, two Dorylaimia (Nematoda: Enoplea) species, *Romanomermis culicivorax* and *Trichuris muris*, were used as outgroups. PhyloSuite was used to abstract the sequences of 36 genes (12 PCGs, 2 rRNAs and 22 tRNAs). All GenBank files were rearranged to start with the *cox1* gene using the *Reorder* function in the same program, and the program was also used to generate the comparative mitogenomic architecture table for *P. sinensis* and *C. robustus*. All subsequent phylogenetic analysis steps were also conducted using PhyloSuite and its plug-in programs. Genes were aligned using MAFFT [38]: the normal alignment mode was appropriate for the rRNAs or tRNAs, and the codon alignment mode was appropriate for the 12 PCGs. Subsequently, a single alignment was generated by concatenating the manually optimized alignments using PhyloSuite. Based on the Akaike information criterion, GTR+G model was selected by PartitionFinder2 [39] as the best-suited evolutionary model. Two different algorithms: maximum likelihood (ML) and Bayesian inference (BI) were used to accomplish phylogenetic analyses. Bayesian inference was performed with default settings in MrBayes 3.2.6 [40] and accompanied with 1 × 10^6^ metropolis-coupled MCMC generations. The best-fit partitioning strategy and models for partitions for IQ-TREE [41] were inferred using the inbuilt functions of IQ-TREE and followed by phylogenetic reconstruction with 50,000 ultrafast replicates [42]. Gene orders were visualized and annotated by iTOL [43] with the help of PhyloSuite. A phylogenetic tree was reconstructed using the MLGO program [44] on the basis of the gene order dataset, with 1000 bootstrap replicates [45].

## 3. Results

### 3.1. Genome Organization and Nucleotide Composition

The circular mitogenome of *P. sinensis* was 13,874 bp in length (Figure 1). The annotated mitogenome has been submitted to GenBank with accession number MW971502. The mitogenome encoded 36 genes: 12 PCGs (*nad1-6*, *nad4L*, *cox1-3*, *atp6* and *cytb*), 22 tRNA genes (one tRNA for each amino acid, apart from leucine and serine, each of which was encoded by two tRNA genes) and 2 rRNA genes (s-rRNA and l-rRNA). All genes are transcribed from the same DNA strand. The absence of the *atp8* gene in accordance with the general features of nematode mitogenomes with the exception of *Trichinella spiralis*, in which the *atp8* gene was found [46]. In comparison to other nematodes [47,48], the base composition of the total mitogenome for *P. sinensis* was biased toward T and G (A + T content of 68.7%; 46.9% T; 22.4% G; 21.8% A; 8.9% C) rather than A and T, but the strand bias is consistent with other nematodes with a negative GC-skew and positive AT-skew. In conclusion, the nucleotides of metazoan mitogenomes are not randomly distributed, and this nucleotide bias is often linked with unequal usage of synonymous codons [49].

### 3.2. Protein-Coding Genes and Codon Usage

The twelve PCGs ranged in size from 234 bp (*nad4L*) to 1584 bp (*nad5*). Among the 12 PCGs, seven (*nad1*, *nad2*, *nad3*, *nad6*, *cytb*, *cox1* and *cox3*) used TTG as the start codon, whereas four (*nad4*, *nad4L*, *nad5*, *atp6*) used ATT, and GTG was used only by the *cox2* gene. TAA was the most commonly used termination codon (*nad3*, *nad4*, *nad4L*, *nad5*, *nad6*, *cox3*, *atp6*, *cytb*); three genes (*cox1*, *cox2*, *nad1*) used TAG, and the incomplete termination codon T-- was inferred only for the *nad2* gene (Table 2). Same as *C*. *robustus*, the PCGs of the *P. sinensis* mitogenome are biased toward using amino acids encoded by T-, G- and A-rich codons. Codon usage, RSCU and codon family proportion (relevant to the amino acid usage) of *P. sinensis* and *C. robustus* are presented in Figure 2. Leucine (17.17%), phenylalanine (14.93%) and serine (10.22%) were the most frequent amino acids in the PCGs of *P. sinensis*, whereas arginine (0.97%), glutamine (1.33%) and histidine (1.5%) were relatively insufficient. The codons TTT (Phenylalanine, 14.3%) and TTG (Leucine, 8.6%) usage were most frequent, while the CGA codon for arginine was scarce (Appendix A). Therefore, codons ending in A or T were predominant, which consistent with the high A + T content of the third coding position of all PCGs in *P. sinensis* (Figure 2). The ratios of dN/dS for all twelve PCGs of *P. sinensis* against *C. robustus* ranged from 0.002 to 0.034 (Figure 3). All the PCGs were under negative (purifying) selection (*ω* < 1) pressure, indicating the existence of functional constraints affecting the evolution of these genes. Functional constraints on *nad3*, *cytb*, *nad1* and *nad6* genes were the most relaxed, whereas *cox2* gene was evolving under the strongest purifying selection pressure (Figure 3).

### 3.3. Transfer RNA and Ribosomal RNA Genes

The small and large subunits of *rrn* (*rrnS* and *rrnL*) of *P. sinensis* were 682 bp and 945 bp in size, with 65.6% and 72.6% AT contents, respectively (Appendix A). All 22 commonly found tRNAs were identified in the mitogenome of *P. sinensis*, ranging from 52 bp (*trnN* (*gtt*)) to 62 bp (*trnM* (*cat*) and *trnK* (*ttt*)) in size, adding up to 1238 bp in length, with an average A + T content of 69.7% (Table 2). All 22 tRNA genes do not exhibit the standard cloverleaf secondary structure. The inferred structures of 20 of the twenty-two tRNA genes (except for two *trnS* genes) in *P. sinensis* consist of an amino-acyl stem of seven nucleotide pairs (ntp), a DHU-stem of 4 ntp with a 3–7 nt loop, an anticodon stem of 5 ntp with a loop, and a unique characteristic in that the TΨC arm and variable loop are replaced by a ‘TV replacement loop’ of 6–14 bases. The *trnS* possesses a secondary structure including a DHU replacement loop of 4 bases, a 5 bp TΨC arm, a TΨC loop of 4 bases and a variable loop of 3–4 bases. Noncanonical tRNA structure are common in nematodes, and secondary structures predicted for the transcripts of these genes are similar to those reported from a variety of other nematode mitogenomes [47,48,50,51,52,53,54], excluding *T. spiralis* in which some tRNAs have the canonical cloverleaf structures [40].

### 3.4. Non-Coding Regions

There were 8 short intergenic regions ranging from 1 to 58 bp in size, adding up to a total of 393 bp. In addition, there were also two long non-coding regions (>100 bp): NCR1 (between *trnS2* and *trnY*) and NCR2 (between *trnV* and *trnP*). They were 284 bp and 417 bp long, with 61.3% and 60.4% AT content, respectively (Appendix A). Tandem repeats are inferred to result from strand slippage during replication [55], and they have been identified in large NCRs of nematodes, including CR1-CR6 in *Caenorhabditis elegans* and the dinucleotide repeat region in *Ascaris suum* [52]. However, the two NCRs of *P. sinensis* were absence of consecutive sequences of A and T, and we did not identify any tandem repeats.

### 3.5. Phylogeny

Phylogenetic relationships of *P. sinensis* with related nematodes of the Spirurina were presumed using ML and BI methods. The dataset comprised 36 concatenated genes (12 PCGs, 2 rRNAs and 22 tRNAs). Both methods, BI and ML, produced an identical topology, and most of the nodes exhibited high bootstrap resampling (ML) and posterior probability (BI) values. Appendix A shows the combined strict accordance phylogenetic tree produced from the two phylogenetic methods. As some superfamilies were represented by a single species in our study, this topology should be comprehended with some caution. Tree topology suggests the existence of two major clades: one clade includes the superfamilies Physalopteroidea, Thelazioidea, Spiruroidea and Filarioidea; another clade consists of the superfamilies Oxyuroidea, Camallanoidea, Dracunculoidea, Rhigonematoidea, Heterakoidea, Seuratoidea, Gnathostomatoidea and Ascaridoidea. The second clade was further sub-divided into two clades, Oxyuroidea and (Camallanoidea + (Dracunculoidea + (Rhigonematoidea + (Heterakoidea + (Seuratoidea + (Gnathostomatoidea + (Ascaridoidea))))))), and a monophyletic clade comprising (Physalopteridae + (Thelazioidea + (Gongylonematidae + (Setariidae + (Onchocercidae))))). Most clades were robustly supported. This relationship is in agreement with phylogenies inferred by Černotíková et al. [56] and Zou et al. [5]. However, our results appear to reject the hypothesis that Oxyuroidea represented the earliest branch of this clade [3,20], our results in agreement with a previous study [57].

The topology obtained in our phylogenetic analysis supports a sister relationship between Seuratoidea and Ascaridoidea, while strongly rejecting the Ascaridoidea-Spiruroidea affinity [47,50]. Meanwhile, the sister relationship also is supported between the Camallanoidea and Dracunculoidea. Ascarididae and Anisakidae were resolved as paraphyletic, along with the particularly problematic phylogeny of the *Ortleppascaris* genus, which appears to exhibit a closer relationship with the family Anisakidae than Ascarididae. Intriguingly, the *Ophidascaris* genus is closer to the family Ascarididae than Anisakidae. In our study, in addition, Gnathostomatoidea formed a sister clade with (Seuratoidea + Ascaridoidea), which is also incongruent with Zou et al. [5]. And the sister group relationship between the Quimperiidae family and Cucullanidae family was revealed, mitochondrial gene order and transformational pathway analysis provided additional supporting evidence for the close relationship between these two families in this study (Appendix A).

### 3.6. GO Analysis

For a better comparison of gene orders among the Spirurina, Phylosuite was used to extract and visualize all available sequences from the NCBI. The gene order of *P. sinensis* is very similar to that of *C. robustus*, with only one transposition event: a position change of *trnN*; three TDRL operations are required to *Gnathostoma doloresi*; a single transposition event and two TDRL operations are required to *Ophidascaris baylisi* (Appendix A). Such gene arrangements are interpreted to be important for molecular systematics and phylogenetic reconstruction because of the rarity of mitochondrial gene arrangement events [58]. Meanwhile, the mitochondrial gene orders of *P. sinensis* and *C. robustus* show that Seuratoidea species do not shows a synapomorphic gene order, and indicate that gene order within this group may be evolving at uneven rates.

In order to evaluate the phylogenetic power of gene order, we reconstructed a phylogeny using gene orders (MLGO algorithm), and compared it with the phylogeny inferred using mitogenomic sequences. The monophyly of (Physalopteroidea + (Spiruroidea + (Thelazioidea + (Filarioidea)))) was supported both by the gene order and mitogenomic sequences. Paraphyly of Camallanoidea and Oxyuroidea was also supported by both topologies, as was the sister-group relationship of *Pingus* and *Cucullanus* genera. There were also some differences: *Onchocerca flexuosa*, *Chandlerella quiscali*, *Litomosoides sigmodontis* and *Ruizia karukerae*, which have gene arrangements that do not conform to the gene orders of their closest relatives and topologies. Disregarding the tRNA genes, there were only 9 types of gene orders; this has been believed as additional evidence of their close relationship [6] (Appendix A), tRNA genes possess small size which probably result in translocations accumulate more quickly. [59]. Hence, designating tRNA genes as hypermobile and excluding them from gene order phylogenies is a feasibile option when dealing with different rearrangement rates [60].

## 4. Discussion

In conclusion, our analysis reveals the phylogenetic position of family Quimperiidae and provides an insight into the phylogenetic relationships of Spirurina, although the availability of molecular data remains limited. We have sequenced and characterized the complete mitogenome of the fish-parasitizing spirurine nematode *P. sinensis*, which includes the standard 22 rRNA genes, 2 rRNA genes and 12 PCGs. There are no highly repetitive regions in the NCR. Using concatenated nucleotide sequences of all 36 genes, the phylogenetic analyses were conducted using Bayesian inference and maximum likelihood methods, providing insight into the relationships within the Spirurina. Meanwhile, gene order can be used with discretion as a source of phylogenetic data in nematodes. The monophyly of (Physalopteroidea + (Spiruroidea + (Thelazioidea + (Filarioidea)))) and paraphyly of Camallanoidea and Oxyuroidea are supported both by the gene order and mitogenomic sequences, as was the sister-group relationship of *Pingus* and *Cucullanus* genera. However, gene orders are inclined to produce artefactual relationships and should be accompanied with utmost caution for phylogenetic reconstruction in Nematoda. Further studies should centre around adding mitogenomes of other inadequately represented taxa for purpose of advancing understanding of the evolution and variation of nematode trophic ecologies and infer their evolutionary relationships with higher resolution.

## Figures and Tables

**Figure 1 genes-12-01772-f001:**
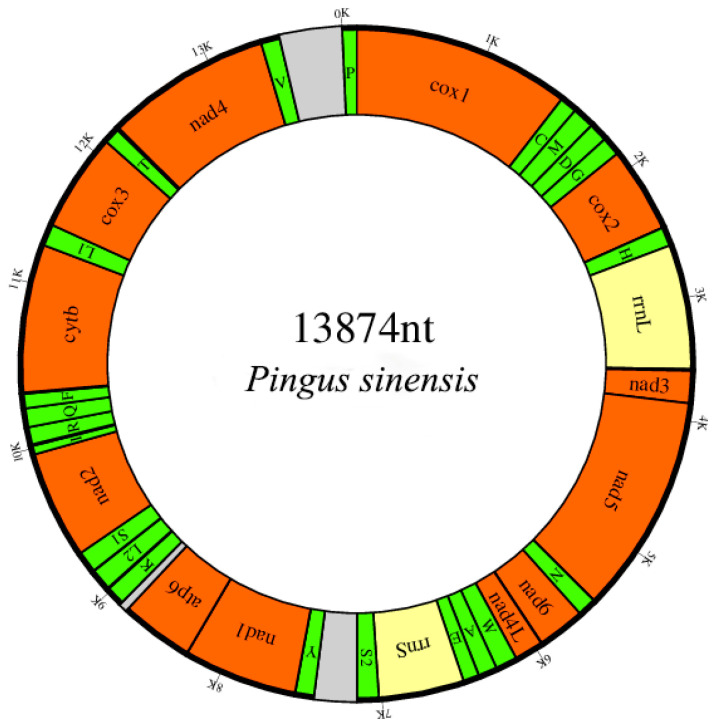
Mitochondrial map of *Pingus sinensis*. The genes are denoted by the color blocks. PCGs are orange, rRNAs are yellow and tRNAs are green. The tRNAs are labelled according to the IUPACIUB single-letter amino acid code, with numerals distinguishing each of the two leucine- and serine-encoding tRNA genes (L1 and L2 for CUN and UUR, respectively; S1 and S2 for AGN and UCN, respectively), and intergenic regions are grey.

**Figure 2 genes-12-01772-f002:**
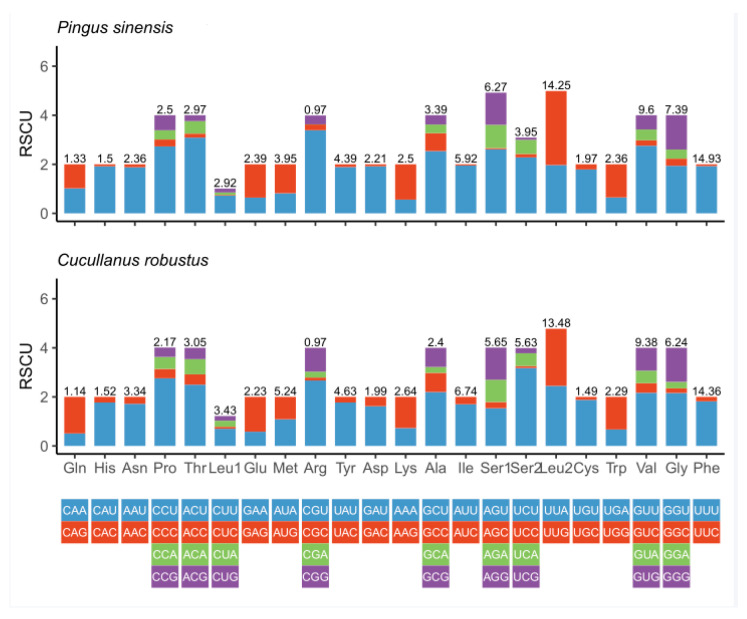
Relative Synonymous Codon Usage (RSCU) of *P. sinensis* and *C. robustus*. Values on the top of the bars refer to amino acid usage.

**Figure 3 genes-12-01772-f003:**
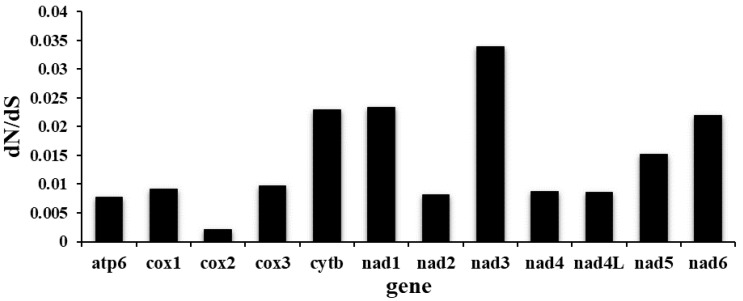
Ratios of non-synonymous (dN)/synonymous (dS) nucleotide substitutions calculated from all 12 PCGs of *P. sinensis* and *C. robustus* mitogenomes.

**Table 1 genes-12-01772-t001:** Primers used to amplify and sequence the mitochondrial genome of *P. sinensis*.

Fragment No.	Gene or Region	Primer Name	Sequence (5′-3′)	Length (bp)
F1	*COX1*	C1F1	ATTGGGGGGTTTGGTAATTG	506
		C1R1	TAAACTTCAGGGTGGCCAAA	
F2	*COX1-16S*	C1F2	TGGCAGGAGCTATTACTATG	2452
		C1R2	CATAAGCCGAAGACTTCATTC	
F3	*16S*	C1F3	ATGGCAGTCTTAGCGTGAGG	404
		C1R3	TCTATCTCGCGATGAATTAAAC	
F4	*16S-ND5*	C1F4	CAGGAAATCATGCTAGATAG	1328
		C1R4	CTACCAAAGCAGAAAATCTAGC	
F5	*ND5*	C1F5	GAAGGTGGTTGCCTAAGGCGA	234
		C1R5	GAGATAAAGTTCTCAAAGCAAC	
F6	*ND5-12S*	C1F6	TGGAGAACTTGTGTTGGTG	2068
		C1R6	GTTTAACAAATCTTAATTCTG	
F7	*12S*	C1F7	TGTTCCAGAATAATCGGCTA	511
		C1R7	CAATTGATGGATGATTTGTACC	
F8	*12S-ND1*	C1F8	TGTGGATCTAGATTAGAG	802
		C1R8	GACAAACCAGGTACTAAC	
F9	*ND1*	C1F9	ACGTTTGGGTCCTAATAAGG	482
		C1R9	CTGAAAAATCAAAAGGCGCC	
F10	*ND1-CYTB*	C1F10	GTGTTGCTTATGAGATTGC	2902
		C1R10	CCAATAACCCCCAAGGTAAC	
F11	*CYTB*	C1F11	GGCTCAAATGAGGTTTTGGGC	412
		C1R11	ATATCACTCAGGAACAATATGG	
F12	*CYTB-ND4*	C1F12	GTTTTGTTGAGGCCCTTTAG	1967
		C1R12	CCTAAAAGAACTACCCAAAAC	
F13	*ND4*	C1F13	GCTCATGTTGAAGCACCTAC	206
		C1R13	GAAGAATAAGCAGCCAAAG	
F14	*ND4-COX1*	C1F14	GTTTTGGGTAGTTCTTTTAGG	1481
		C1R14	GTACCACAACCCAAATCAAC	

**Table 2 genes-12-01772-t002:** The genome organization of *P. sinensis*.

	Position				Codon		
Gene	From	To	Size	Intergenic Nucleotides	Start	Stop	Anticodon
*trnP*	1	57	57				TGG
*cox1*	58	1623	1566		TTG	TAG	
*trnC*	1623	1678	56	−1			GCA
*trnM*	1679	1740	62				CAT
*trnD*	1742	1798	57	1			GTC
*trnG*	1799	1853	55				TCC
*cox2*	1854	2540	687		GTG	TAG	
*trnH*	2539	2593	55	−2			GTG
*rrnL*	2594	3538	945				
*nad3*	3551	3877	327	12	TTG	TAA	
*nad5*	3877	5460	1584	−1	ATT	TAA	
*trnN*	5459	5510	52	−2			GTT
*nad6*	5511	5939	429		TTG	TAA	
*nad4L*	5944	6177	234	4	ATT	TAA	
*trnW*	6177	6232	56	−1			TCA
*trnA*	6233	6288	56				TGC
*trnE*	6288	6342	55	−1			TTC
*rrnS*	6343	7024	682				
*trnS2*	7025	7078	54				TGA
*NCR1*	7079	7362	284				
*trnY*	7363	7420	58				TAC
*nad1*	7419	8291	873	−2	TTG	TAG	
*atp6*	8294	8890	597	2	ATT	TAA	
*trnK*	8949	9010	62	58			TTT
*trnL2*	9018	9073	56	7			TAA
*trnS1*	9071	9128	58	−3			TCT
*nad2*	9129	9960	832		TTG	T	
*trnI*	9961	10016	56				GAT
*trnR*	10027	10080	54	10			ACG
*trnQ*	10081	10134	54				TTG
*trnF*	10135	10191	57				GAA
*cytb*	10183	11292	1110	−9	TTG	TAA	
*trnL1*	11293	11350	58				TAG
*cox3*	11351	12118	768		TTG	TAA	
*trnT*	12119	12174	56				TGT
*nad4*	12190	13404	1215	15	ATT	TAA	
*trnV*	13404	13457	54	−1			TAC
*NCR2*	13458	13874	417				

## Data Availability

Not applicable.

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
