# Peer review of "Sequencing of the Complete Mitochondrial Genome of Pingus sinensis (Spirurina: Quimperiidae): Gene Arrangements and Phylogenetic Implications"

_genes, 2021, doi:10.3390/genes12111772_

Round 1

Reviewer 1 Report

This paper documents the complete sequence of the mitochondrial genome from Pingus sinensis, a snakehead parasite that is similar to other nematodes.  The authors determine the complete 13,874 bp DNA sequence and assign the gene organization.  The authors examine the codon usage of the 12 genes and relate these to Cucullanus robustus.  A phylogenetic analysis was performed using Bayesian inference and GO analysis to determine the evolutionary relationship to other nematode mtDNA genomes.  The work is straightforward and solid.

Comments:

It would benefit the paper to further examine the non-coding regions, especially in terms of how veracious conserved regions may participate in mtDNA replication and transcription initiation.  How does this NCR relate to other nematodes, yeast and mammalian mtDNA genomes.  Can the authors determine any critical regions important for replication?  The following papers may offer some guidance.

Lewis, S.C. et al., (2015) A Rolling Circle Replication Mechanism Produces Multimeric Lariats of Mitochondrial DNA in Caenorhabditis elegans. PLoS Genet 11(2): e1004985. doi:10.1371/journal.pgen.1004985

Lemire B (2005) Mitochondrial genetics. WormBook: 1–10. doi: 10.1895/wormbook.1.25.1.

Author Response

Dear Reviewer,

Thank you very much for the time you have invested in reviewing our manuscript and references recommended. In the mitogenome of Xiphinema Americanum, a sequence motif of an NCR was found and thought to act as the promoter for light strand transcription, furthermore, a stem-loop structure inferred for the Heavy strand which contains long stretches of T in the loop region was predicted as the initiation site of light strand replication. And a second long NCR has been found in Caenorhabditis elegans and Ascaris suum mitogenomes, which contains a stem-loop structure resembling the replication initiation site of the light strand. Based on these, we suspect that similar sequence motifs serve as the initiation site of light strand replication of Pingus sinensis mtDNA, of course, this needs to be further verified, and we have revised the manuscript according to the comments and the recommendations suggested by the editors and reviewers. Thank you again for your great help and attention, your comments are valuable and important guiding significance to our further research. I am looking forward to hearing from you about the final decision.

Best regards and wishes!

Yours sincerely,

Fang-Lin Chen, Hong Zou, Xiao Jin, Dong Zhang, Wen-Xiang Li, Ming Li, Shan-Gong Wu, Gui-Tang Wang

Reviewer 2 Report

The work presented in this manuscript is well written and provides insights to a topic that is still missing scientific information. However, I am a bit puzzled about the approach used, even though a similar paper by the corresponding author was published a few years ago. Primer walking used to be a valid approach for "non-standard" genomes, though second and third generation sequencing and recent  bioinformatics tools can prove to be valuable in resolving such genomes without the need of "manual" processing. In this regard, I would have liked to know more about how the mitogenome was "assembled manually" as this may lead to unintentionally biased results. I would suggest the authors to perform some high-throughput sequencing (e.g. Illumina) and map the reads back to the assembled mitogenome to verify that this is a proper representation. Considering that the authors regard the size and gene order of this mitogenome as important, I believe this is important to consider.

Author Response

Dear Reviewer,

Many thanks for your high efficiency of work and good suggestions. In contrast to primer walking, high-throughput sequencing results in longer read length at low cost and we also have read some papers about this, but primer walking is well established in our lab. And the most important reason is that we cannot conduct high-throughput sequencing due to the limitations of sample size and quantity. In comparison to that, primer walking is able to achieve the desired purpose and resolve such problems serving as a valid approach. All obtained fragments were quality-proofed (electropherogram) and BLASTed to confirm that the amolicon is the actual target sequence. Whenever the quality was sub-optimal, sequencing was repeated. All obtained fragments were BlASTed to confirm that the amplicon is the target sequence. Mitogenome was assembled stepwise with the help of DNAstar program, making sure that the overlaps were identical, and that no NUMTS were incorporated into the sequence, and we have revised the manuscript according to the comments and the recommendations suggested by the editors and reviewers.

Thank you again for your great help and attention, your comments are valuable and important guiding significance to our further research and urge us to seriously improve our research methods. I am looking forward to hearing from you about the final decision.

Best regards and wishes!

Yours sincerely,

Fang-Lin Chen, Hong Zou, Xiao Jin, Dong Zhang, Wen-Xiang Li, Ming Li, Shan-Gong Wu, Gui-Tang Wang
